# VISUAL EXPERTISE AND THE LOG-POLAR TRANSFORM EXPLAIN IMAGE INVERSION EFFECTS

## ABSTRACT

Visual expertise can be defined as the ability to discriminate among subordinate-level objects in homogeneous classes, such as identities of faces within the class "face". Despite being able to discriminate many faces, subjects perform poorly at recognizing even familiar faces once inverted. This face-inversion effect is in contrast to subjects' performance identifying inverted objects for which their experience is at a basic level, which results in less impairment. Experimental results have suggested that when identifying mono-oriented objects, such as cars, car novices' performance is between that of faces and other objects. We build an anatomically-inspired neurocomputational model to explore this effect. Our model includes a foveated retina and the log-polar mapping from the visual field to V1. This transformation causes changes in scale to appear as horizontal translations, leading to scale equivariance. Rotation is similarly equivariant, leading to vertical translations. When fed into a standard convolutional network, this provides rotation and scale invariance. It may be surprising that a rotation-invariant network shows any inversion effect at all. This is because there is a crucial topological difference between scale and rotation: Rotational invariance is discontinuous, with V1 ranging from 90°(vertically up) to 270°(vertically down). Hence when a face is inverted, the configural information in the face is disrupted while feature information is relatively unaffected. We show that the inversion effect arises as a result of visual expertise, where configural information becomes relevant as more identities are learned at the subordinate level. Our model matches the classic result: faces suffer more from inversion than mono-oriented objects, which are more disrupted than non-mono-oriented objects when objects are only familiar at a basic level.

## 1 INTRODUCTION

Since 1969, researchers have been studying the effects of inverting images (Yin, 1969). Some researchers have focused on defining the bounds of inversion effects: what the measurable effect is for what types of images (Farah et al., 1995; Yin, 1969; Jacques et al., 2007; Rezlescu et al., 2017). Others looked to explain how inversion effects arise: what part of the brain was active during inversion tasks or what level of experience a participant had with the stimuli in the experiment (Gauthier et al., 2000; Gauthier & Bukach, 2007; Gauthier et al., 2014; Kanwisher et al., 1997; 1998; Richler et al., 2011; Wang et al., 2014).

In Yin (1969), participants studied a set of images during the training phase, and then they were shown pairs of images in testing and asked to select the image that was in the study set. Trials with upright images and trials with inverted images were compared to determine the inversion effect. Using images of faces resulted in a strong and significant inversion effect - performance was much worse for inverted faces. Images of houses - a mono-oriented category - had a lesser, but still significant effect. Images of airplanes had an insignificant inversion effect. We draw two conclusions from this work: the effects of inversion on performance are greater when images of faces are used as the stimuli and insignificant when images of certain objects (e.g., planes, that are less mono-oriented than houses) are used as the stimuli (Yin, 1969). The second conclusion is that not all objects produce the same inversion effects. Mono-oriented objects, which are objects that are typically seen in only one orientation such as the houses in Yin's 1969 work, do show an inversion effect, though it is smaller than that of faces (Yin, 1969).

Since Yin's 1969 paper on inversion, a great deal of research has focused on explaining inversion effects. Why is it that different stimuli - faces, objects, mono-oriented objects - produce different inversion effects? One explanation of inversion effects, supported by brain imaging and behavioral studies, is that visual expertise changes the way we process visual stimuli. Faces are processed holistically, which means that not just the features, but the *configuration* of the features matters. When such stimuli are inverted, the configuration is disrupted, and we are left with featural processing (Gauthier et al., 2000; 1999; 2003). Similar inversion effects have been observed in experts of other domains, such as dog show judges or bird watchers (Diamond & Carey, 1986; Gauthier et al., 2000).

Visual expertise is defined with respect to Rosch's basic level categories. In a category hierarchy, the basic level is the level at which objects are most commonly labeled, such as "chair", "tree", or "car". Basic level categories define broad categories of objects that share properties such as general appearance, function, and common parts (Rosch et al., 1976). For example, cars can look very different from each other, but they all have wheels, an enclosed space for passengers, and are used for ground transportation. Visual expertise is defined as having proficiency in differentiating subordinate-level sub-classes of basic level categories. For example, subordinates of the basic level category "tree" could include "sugar maple", "american elm", or "northern red oak".

Most people are face experts in this sense. It has been estimated that we are able to identify on the order of 5,000 different people (Jenkins et al., 2018). Identity is a subordinate-level judgment because faces share the same features (eyes, nose, mouth, ears, etc.) in the same general configuration. We also process faces holistically (Gauthier & Bukach, 2007). This means that instead of just using the features of a face to recognize a person, we use the configural information, such as the distance between the eyes, or the distance from the nose to the mouth. Hence, expertise is fine grained discrimination of homogeneous categories. The research into expertise suggests that experts in other domains, such as cars or birds, also use configural information when viewing basic level categories in which the participants are experts (Gauthier et al., 2000).

We conduct experiments in order to ask if there is a way to characterize inversion effects in different stimuli in terms of levels of expertise. In doing so, we explore the changes in visual signal processing that occur between novice level and expert level. To do this, We build an anatomically inspired network that incorporates foveation - high resolution central vision and low resolution peripheral vision - and the log polar mapping between the visual field and the primary visual cortex (Polimeni et al., 2006). The log polar mapping causes changes in image scale to appear as horizontal translations. When presented as input to a convolutional neural network (CNN), which is translation invariant, the log polar mapping makes the network relatively scale invariant. Image rotation is similarly equivariant in the log-polar representation, because it leads to vertical translations. However, the two differ topologically: pixels that shift vertically can "fall off" the edge of the image and wrap around to the opposite edge. This causes a rearranging of features in the image, hence a disruption of configural information.

Using this model, we test the inversion effects of different types of stimuli across increasing expertise in order to gain an insight into how and why visual processing changes based on the visual stimulus. Our model is consistent with the view that expertise plays a significant role in the way we process visual inputs, and leads to the inversion effects seen in previous work.

## 2 METHODS

### 2.1 MODEL

We use ResNet-50 (He et al., 2016) to perform all experiments, trained from scratch with the foveated, log-polar representation. We call this LPnet. We compare our results to a "vanilla" ResNet-50 with standard images. Unless otherwise noted, all experiments use the Adam optimizer, an initial learning rate of 1e-4, and a minibatch size of 48.

### 2.2 DATA

To test the effects of expertise in visual processing, we use four different datasets. To model experts, the first three datasets are images of faces, cars, and dogs, generally mono-oriented objects, with targets at the subordinate level. To model novices, who mainly know basic-level labels, the fourth

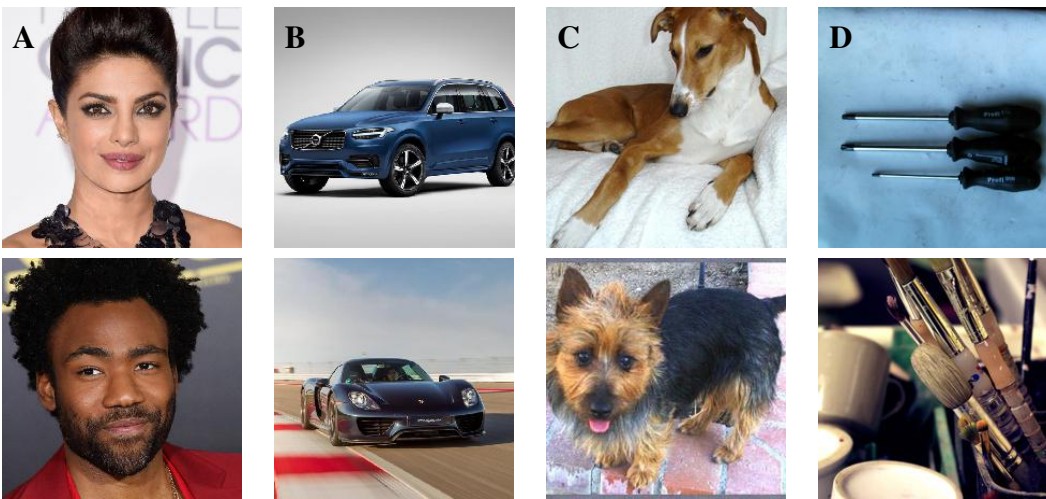

Figure 1: Example images used for recognition experiments from four datasets: (A) a lab-gathered face dataset, (B) Comprehensive Cars dataset, (C) ImageNet (dog categories only), and (D) ImageNet (dog categories excluded).

dataset contains 128 categories with basic-level targets. 124 are ImageNet categories, containing objects that are seen in a variety of orientations in natural scenes. The remaining four are mono-oriented: faces, cars, houses, and dogs. We randomly chose our sets with a 80/10/10 split.

**Faces** We used a dataset collected by lab members which includes 128 separate identities and approximately 200 example images per identity. The images for each identity portray the person in a variety of contexts, with differing backgrounds, lighting conditions, orientations, and facial expressions. Example images from the dataset are shown in Figure 1A.

**Subordinate level mono-oriented objects: Cars** Cars are an appropriate choice for mono-oriented objects because they are almost exclusively seen upright in natural scenes. "Car" is also a basic level category with a number of sub-classes. Car expertise is associated with discrimination at the level of model, e.g., 2010 Toyota Camry. The dataset used is the Comprehensive Cars dataset (Yang et al., 2015). These images include cars of a variety of makes, models, and years. The dataset contains 136,726 total images across 163 car makes and 1,716 car models. The cars are of varying orientations, lighting conditions, and backgrounds. Examples of cars from the Comprehensive Cars dataset are shown in Figure 1B.

**Subordinate level mono-oriented objects: Dogs** In the same way that models of cars are subordinates of the category "car", dog breeds are subordinates of the category "dog". We use 117 dog breeds included in ImageNet (Deng et al., 2009). Like all of ImageNet, these images are highly variable in pose, context, and scale. Because of this we use all available images from ImageNet in these 117 categories. Examples of dogs in the dataset are shown in Figure 1C.

**Basic level categories** We use a subset of 124 categories from ImageNet (Deng et al., 2009) which were chosen specifically because they are naturally viewed in multiple orientations, such as "ladle", "screwdriver", or "dumbbell" (Figure 1D). We avoided categories such as "clock" or "candle" which, although can be oriented multiple ways, are naturally seen primarily in a limited number of orientations. The labels for the stimuli in this category are at the basic level, instead of at the subordinate level as in the previous two datasets (Rosch et al., 1976). Again, due to the within-category variance, we used all examples in ImageNet in order to achieve acceptable performance. We also included 4 categories (bringing the total number of categories to 128) that are mono-oriented. They are: "face", "car", "dog", and "house". Using these mono-oriented objects in our experiment with basic level categorization allows us to compare how performance changes between experts and novices as visual expertise general increases. We aggregate data from each of the subordinates of "face", "car", and "dog" to get a varied sample of images for these categories. We matched the number of images approximately to the number of images the ImageNet categories included.

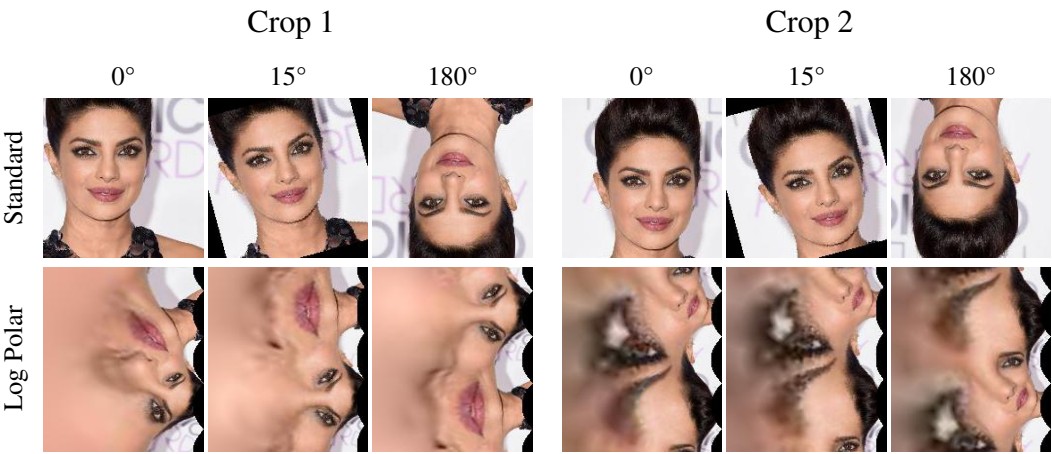

Figure 2: A visualization of the log polar transformation for two crops of the same image. The top row shows standard images, or a depiction of the visual field. The bottom row shows images that have been transformed, using the log polar mapping that approximates the representation of the visual field in V1. For each crop, we show both image types at three amounts of rotation: 0°, 15°, and 180°.

## 2.3 DATA TRANSFORMATIONS

**Cropping** We augment our data by performing random cropping on the images. We take four random crops of each image, with each crop including approximately 65% of the pixels in the image. The crops cannot extend past the edge of the image, so none of the crops include any padding.

**Rotation** We randomly rotated our training images to be between -15°and 15°, because scenes may be viewed with some small amount of rotation from the tilting of the viewers own head. To study the effects of inversion in the network, our validation images are shown at 0°and 180°.

**Foveation** For LPNet only, we foveate each crop using the algorithm described in Jiang et al. (2015). The foveation leaves the center of the crop (the point of fixation) at a high resolution and transforms the periphery to be at a lower resolution. The further a pixel is from the center of the image, the greater the degree of blurring. This mimics the foveation of the retina (Jiang et al., 2015).

**Log-polar transform** For LPNet only, we further preprocessed our images to create an anatomically-realistic mapping of the visual field onto the visual cortex. Previous work has shown the validity of using log polar transformations as a 2D approximation to the mapping of the visual field onto the visual cortex in primates (Polimeni et al., 2006) and in computational models of facial recognition (Anonymous). For each case of rotation, we first rotated the image and then took the log polar transformation of the image. After the images have undergone a log polar transformation, the changes in degrees of rotation appear as changes in vertical translation. This is in contrast to the shifts that occur when scaled images undergo a log polar transformation, which are in the horizontal direction. This vertical shift causes pixels to "fall off" the edge of V1. Those pixels wrap around to the opposite side of V1. Instead of appearing as a simple translation, changes in rotation in images that have undergone a log polar transformation result in a fundamental rearranging of image components. A visualization of the log polar transform is provided in Figure 2.

All transformations cropping, rotation, and foveation and log-polarization for LPNet are done at the beginning of every epoch. For LPNet, the cropping has a large effect, as the fixation point changes with each crop. To see this, note the difference in the two fixations in Figure 2.

## 3 EXPERIMENT I: FACES AND OBJECTS

Based on Yin (1969), we first explore the difference in the effect of inversion based on whether a participant is viewing faces (objects of expertise) or objects for which the subject is a novice, so only categorized at the basic level. In that work, Yin found that images of faces produced a significant and large inversion effect, while images of airplanes did not produce a significant inversion effect.

## 3.1 EXPERIMENTAL SETUP

We define expertise in our model as the ability to differentiate between subordinates of basic level categories. When using subordinate level visual stimuli, such as individual face identities, the network has to learn to discriminate between very similar visual stimuli. For example, faces share features and the same overall organization of features. Just as with humans, being able to discriminate very few subordinates demonstrates a low level of expertise with a particular basic level category of visual stimuli. Being able to discriminate visually between many subordinates demonstrates high visual expertise with that category. Our network learned to perform categorization tasks using 4, 8, 16, 32, 64, or 128 categories of either faces or non mono-oriented objects.

Over training, we increase the number of classifications the network makes in order to mimic the increase of visual expertise over time, similar to first knowing only family members, then adding family friends, then pre-K, etc. The network is trained for 40 epochs with 4 category outputs (identities for faces, object categories for the "novice" network). After 40 epochs, 4 new categories are added. There is an immediate sharp drop in accuracy, but during the next 40 epochs the network learns the 8 categories. This continues for a total of 240 epochs, across 4, 8, 16, 32, 64, and 128 category outputs. Each phase of training is essentially performing pretraining of the network for the next phase of training by learning features that are helpful for discrimination.

During training, cropped images are rotated randomly between -15°and 15°. Testing images are presented in two conditions: upright (0°) and inverted (180°). The upright condition provides a baseline so that we can measure the amount the accuracy decreases after inversion. The amount the accuracy drops in the inverted condition is relative to the accuracy of the upright condition, so we report the percent drop in accuracy due to inversion, i.e.,

$$Acc_{lost} = (Acc_{up} - Acc_{inv})/Acc_{up} \tag{1}$$

By looking at the percent drop between the accuracies for the two testing conditions, we can determine the effect inversion had on the network's recognition capabilities.

This experiment includes four configurations of networks and data: (1) CNN with face dataset (2) CNN with object dataset (3) LPNet with face dataset and (4) LPNet with object dataset.

## 3.2 RESULTS

We ran all experiments five times and averaged the results. Figure 3 shows the accuracy over training. The first row is for standard CNNs and the second row is LPNet. The effect of the stimuli and the category level of the stimuli, either basic level or subordinate level, is clear. The green line is the training accuracy, the yellow line is the validation accuracy for upright images, and the magenta line is the validation accuracy for inverted images. For the face stimuli, even in early phases of training, there is a performance gap between the upright validation accuracy and the inverted validation accuracy. As the network learns to differentiate more identities, this performance gap continues to increase. When using objects as the training stimuli, the difference in performance between the two validation conditions is overall much smaller, with no apparent gap during the first phase of training. In addition, the size of the gap changes less throughout training.

The results on the CNN and LPNet have similar trends in that both show performance gaps between the two validation conditions which increase as the number of identities increases. This gap is larger for the CNN, in part because the inverted condition is more difficult for it. With inverted faces, the CNN fails nearly completely on inverted images of faces with an accuracy of approximately 4%. This is because the log polar transform provides some rotation invariance that standard CNNs do not have. Hence LPnet is more representative of human ability on inverted images.

Figure 4 shows the percent accuracy lost as in Eq. 1, i.e., the accuracy lost *relative* to the upright accuracy, directly measuring how much inverting an image will disrupt performance in any given experiment. In Figure 4, it is clear that image inversion has a significant impact with face stimuli, and a much smaller impact with object stimuli. This mirrors the first conclusion of Yin 1969. One of the categories included in the object experiment is houses. In order to recreate Yin's experiment, we also plot the percent accuracy lost on just the house category. The percent accuracy lost for houses is much lower than that of faces, but slightly higher than objects. This mirrors the second conclusion of Yin 1969. Figure 4 also shows how the percent accuracy lost on a standard CNN is higher than the

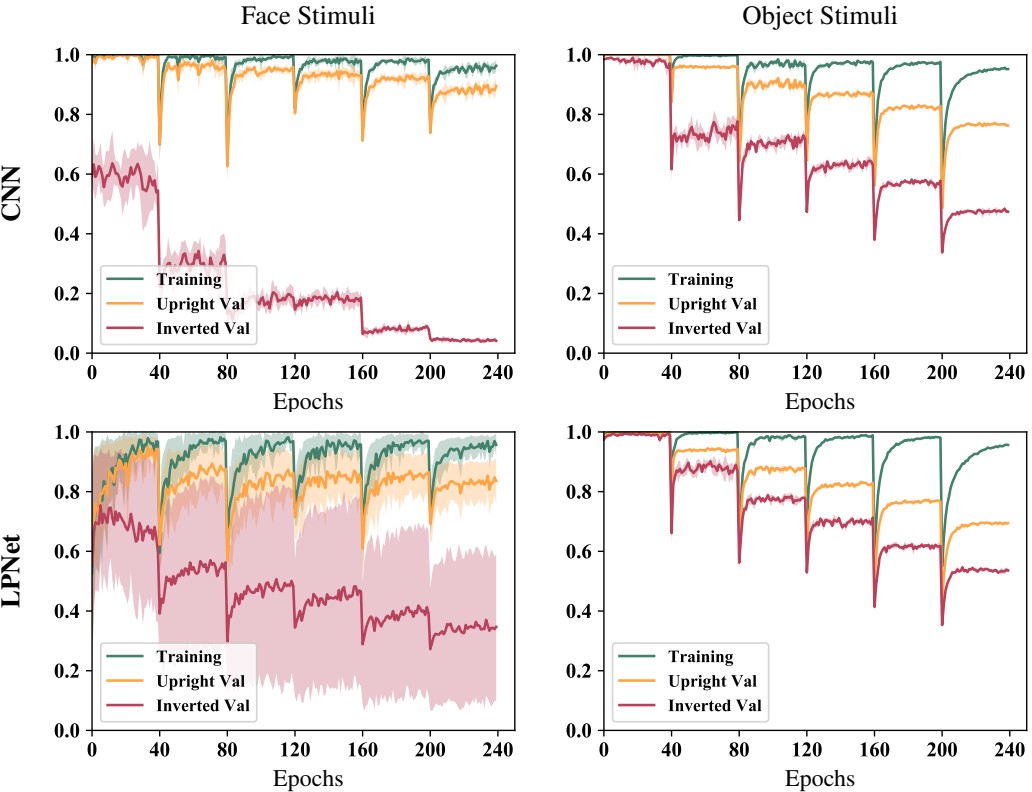

Figure 3: Accuracy throughout training on standard CNNs and LPNet for face and object stimuli. The green line is training accuracy, the orange line is accuracy on validation images at 0°, and the magenta line is accuracy on validation images rotated 180°. Shaded regions are +/-1 standard deviation.

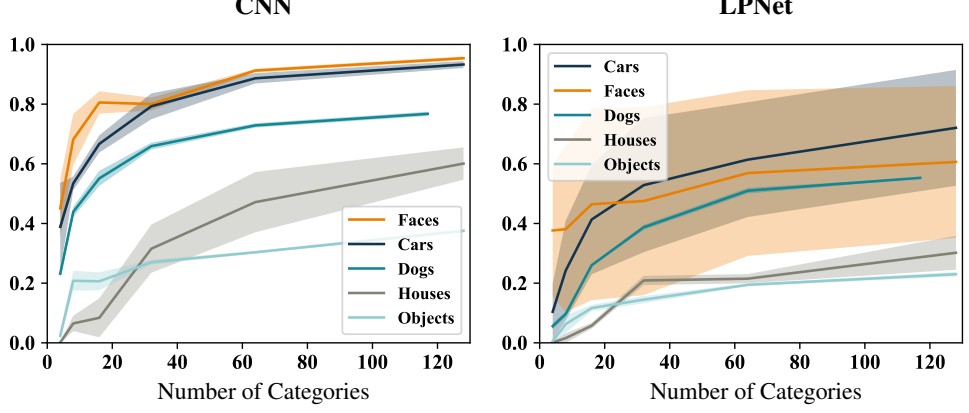

Figure 4: Percent accuracy lost across all phases of training on a standard CNN and LPNet. Shown for faces, car models, and dog breeds (all expert networks), and houses and objects from the basic-level categorizer.

corresponding values on LPnet. This is because there is such a significant performance decrease on inverted images; the standard CNN is losing a larger proportion of the upright validation accuracy.

## 4 EXPERIMENT II: EXPERTISE EFFECTS WITH CARS MODELS AND DOG BREEDS

In the previous experiment we saw that the inversion effect for faces was significant and the inversion effect for objects using a network with ability at the basic level was not significant. Here we explore

expertise with mono-oriented objects, cars and dogs, to compare the inversion effect of faces to that of mono-oriented objects identified by models with expert level knowledge. We can also determine if the degree of expertise of the network changes the magnitude of inversion effects.

## 4.1 EXPERIMENT SETUP

To study expertise in mono-oriented objects, we trained two different sets of networks: one to distinguish the sub-class of car models and one to distinguish the sub-class of dog breeds. Being able to differentiate between different car models (e.g. Toyota Camry, Hyundai Santa Fe) is indicative of a car "expert". We again run experiments with both standard CNNs and LPNet.

Aside from the data chosen, this experiment follows the same setup and procedure as the previous experiment. We increase the number of identities being differentiated during each phase of training. For car models, our phases of training include 4, 8, 16, 32, 64, and 128 category outputs. The mean number of examples per car model used in our experiments is 151 images. For dogs, our phases of training include 4, 8, 16, 32, 64, and 117 category outputs. This is limited by the number of dog breeds available in ImageNet. These images inherently have more variation in pose (dogs sitting, standing, jumping, rolling around, etc.), so we used all available images from the dog categories of ImageNet, which averages to approximately 1500 images per dog breed.

## 4.2 RESULTS

The results for these experiments are shown in Figure 5 with the first row of plots representing results on a standard CNN and the second row representing results on LPNet. For all plots, the validation accuracy gap is much smaller at the beginning of training, but continues to increase as more categories are added. Like the previous experiment, it is much more difficult for the CNN to distinguish inverted images than it is for LPNet, because of the rotation invariance provided by LPNet. Looking just at LPNet plots, the car and dog upright validation accuracies (Figure 5) and the face upright validation accuracies (Figure 3) vary widely. In addition, the number of percentage points lost because of inversion in these experiments also varies. This is because of differences in data that are hard to control for, like more complicated backgrounds for cars and dogs, similar contexts for faces, and amount of data available to train on. Despite these differences, when looking at the percent accuracy lost in Figure 4, it is clear that the network experienced a very similar inversion effect in each expert LPNet network. This is because percent accuracy lost measures how much of the upright validation accuracy was lost due to inversion (how big of an impact inversion had), not the net number of percentage points lost.

Discriminating between different subordinate categories may become harder or easier for the network depending on the stimulus class itself or the context of the images. However, the percent accuracy lost shows that image inversion affects LPNet trained as an expert on car models or dogs in almost the same way it affects LPNet trained on faces. This is consistent with the Diamond & Carey (1986) results.

# 5 EXPERIMENT III: RECREATION OF HUMAN EXPERIMENT: COMPLETE COMPOSITE PARADIGM

To further compare network performance more directly to human data, we recreate the complete composite paradigm (Gauthier & Bukach, 2007; Meinhardt et al., 2014) for LPNet with face data and for a standard CNN with face data. The complete composite paradigm is a way to measure holistic face processing. Subjects view two faces and are asked to attend only to the top half of the face. They are then asked to determine if the top halves are the same or different. This is done in four conditions. In *congruent* trials the answer for the top and bottom images are the same: both the top and bottom halves are the same or both the top and bottom halves are different. Incongruent trials have the same top half of the face with different bottom halves of the face or vice versa. Holistic processing is measured as the difference between congruent and incongruent trials. Figure 6A shows a diagram of the complete composite paradigm from (Meinhardt et al., 2014) and Figure 6B shows example images used in our experiment.

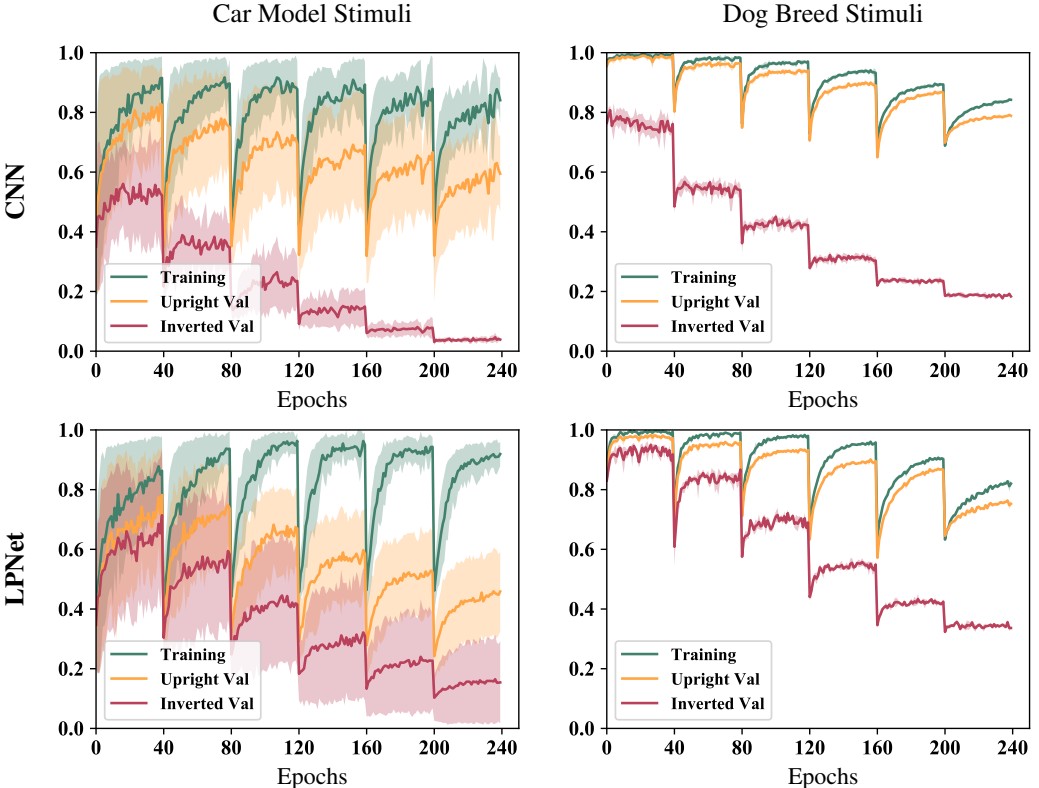

Figure 5: Accuracy throughout training on standard CNNs and LPNet for car model stimuli and dog breed stimuli. LPNet dog breed plot averaged over three runs.

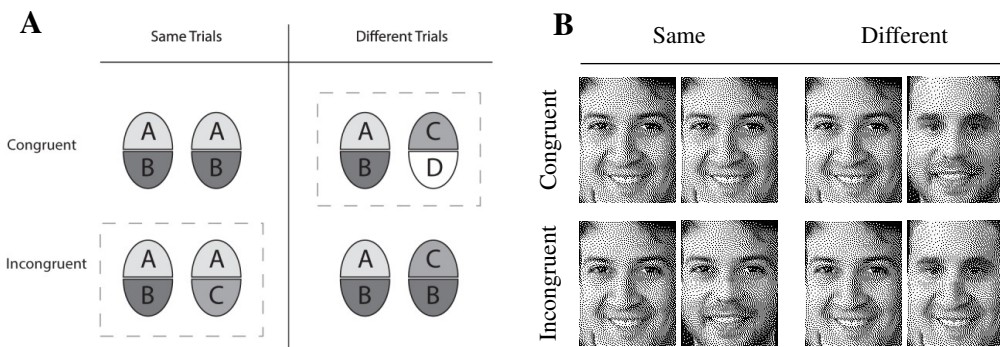

Figure 6: (A) A figure from (Meinhardt et al., 2014) showing different combinations of facial components included in the complete composite paradigm. (B) Images of whole and mismatched faces used to recreate the complete composite paradigm in our network. Here, "Same" judgments are on the top half of the face.

We compare the output from the last convolutional layer of the network for different images and calculate the cosine similarity for a pair of images. The fixation point for the network on the images is between the eyes (as subjects are instructed to attend to the top potion of the face) and three pixels left of center, which comes from data that the first fixation people make when viewing a face is to the left of center (Hsiao & Cottrell, 2008). We perform three trials for each network using three sets of images and average over the trials to get the final cosine similarities. This experiment is done for both upright and inverted images.

| Network | Trial type | Upright Images $\lvert$congruent - incongruent$\rvert$ | Inverted Images $\lvert$congruent - incongruent$\rvert$ |
|---|---|---|---|
| CNN | Same | 0.3086 | 0.4061 |
| CNN | Different | 0.0913 | 0.0661 |
| LPNet | Same | 0.2463 | 0.1983 |
| LPNet | Different | 0.0848 | 0.1196 |

Table 1: Difference in cosine similarity between congruent and incongruent trials for CNN and LPNet.

## 5.1 RESULTS

In Gauthier (2007), the holistic processing effect is calculated as the difference between the congruent and incongruent trials for d' or the hit rate. Because we do not have the data to calculate a hit rate, we instead subtract our cosine similarities between congruent and incongruent trials. As seen in Table 1, both LPNet and the CNN show holistic processing of upright faces with a difference between congruent and incongruent trials being larger than zero. The CNN shows a larger effect, meaning it preforms holistic processing to a greater degree. This helps explain why CNNs preform so poorly on inverted images; they rely on the configural information but do not have any rotation invariance. *The CNN has the same performance with inverted faces.* LPNet, however, sees a drastic decrease in holistic processing for inverted faces, just as people do [REF].

## 6 CONCLUSION

We explored image inversion effects and the impact of visual expertise on performance. By using images of faces, objects, cars, and dogs, we were able to show that LPNet, a convolutional neural network that includes a foveated retina and the log polar mapping from the visual field to V1, can reproduce experimental results of image inversion despite being nominally rotation invariant. We showed that there is a larger effect on performance from inverting images of faces than images of objects, which increases as the number of categories being discriminated increases. We then explored the result that images of mono-oriented objects have an inversion effect between that of faces and objects by showing performance of a house-novice network. Expert networks on mono-oreinted objects do not show an effect between that of faces and objects. We showed that LPNet trained to distinguish car models or dog breeds showed similar inversion effects to faces. This suggests that inversion effects are not dependent on the stimulus, but rather on the level of visual expertise with the stimulus and on the categorization level of the stimulus (expert or novice viewing subordinate or basic level categories).

Image inversion effects are dependent on the level of expertise because of an expert's reliance on configural information to do fine grain discrimination. These effects occur with both CNNs and LPNet. LPNet is more realistic to human vision for the inclusion of the log polar transform, which provides scale and rotation invariance. This is seen with inverted images, when CNNs nearly completely fail to perform differentiation while LPNet is still able to perform some differentiation. We have shown that, when using the log polar transform, the inversion of an image causes a rearrangement of features and a loss of configural information.

The log polar transform approximates the mapping from the visual field to V1, meaning that disruption of configural information occurs before V1 - before the cortex gets the visual input. When LPnet is not an expert, the loss of configural information has a minimal impact on discrimination ability, and it is able to perform the task with only minor performance changes. As LPNet is asked to do more fine grain discrimination between categories, it relies more heavily on the configural information held within the image.

Our results support the hypothesis that the source of the inversion effect in visual expertise, including face expertise, is disruption of configural information at the level of V1.

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
