# OpenReview forum: "Visual Expertise and the Log-Polar Transform Explain Image Inversion Effects"
_ICLR.cc/2023/Conference — Submitted to ICLR 2023_

### Official Review · Reviewer_UAVL · 2022-10-24

**Confidence:** 4
**Correctness:** 3
**Technical Novelty And Significance:** 3
**Empirical Novelty And Significance:** 4
**Recommendation:** 3

**Clarity, Quality, Novelty And Reproducibility:**

- The paper is clear and nicely writtem.

- The paper has quality. However, as explained in the previous section, it still have some clear flaws.

- The paper is novel, as it is the first one looking at inversion effects on CNNs.

- Authors need to provide some better details, specially in their network that includes the foveated rendering and the log-polar transform.

**Strength And Weaknesses:**

Strengths:

- The paper is of interest, as it shows how to modify a CNN in order to perform image inversion in a similar eway to humans. If we aim at creating neural nets that ressemble most to humans, these are lines to follow.
- The paper is very nicely written, and easy to read.

Weaknesses:
-First, I am surprised that the authors completely ignore in the paper that these ideas of looking wether a CNN performs as humans in different tasks has been around for some time. Clear examples of this are [1,2] regarding color and brightness illusions, the Scilliation rig [3], the perception on liquids [4], or contrast sentitivity functions [5]. I believe the authors need a state-of.the-art section including all this line of research.

- The authors perform all the experiments in ResNet-50. I believe that, for a paper as this one, at least a couple of different neural networks architectures should be used. If not, how can we sure that this results do not only hold true for the ResNet architecture.

- The authors in page 2 say that CNN are translational invariant. This is currently not fully agreed by the community, as recent studies have shown that this is not the case. See [6]. For this reason, authors need to comment on this fact.

- The last experiment is not clear. Why cosine similarity? Why not to use any other metric? Can the authors ellaborate?

- Finally, I recommend the authors to remove the last line of the paper. I do not agree that the inversion effect happens in V1. They prove that a log-polar transformation helps, that is suppose to be in V1; however, there is still a long way to claim for that. What happens for example if we modify the architecture.

[1] Convolutional neural networks can be deceived by visual illusions
A Gomez-Villa, A Martin, J Vazquez-Corral, M Bertalmío
CVPR, 2019

[2] Color illusions also deceive CNNs for low-level vision tasks: Analysis and implications
A Gomez-Villa, A Martín, J Vazquez-Corral, M Bertalmío, J Malo
Vision Research 176, 156-174, 2021

[3] ImageNet-trained deep neural networks exhibit illusion-like response to the Scintillating grid
ED Sun, R Dekel - Journal of Vision, 2021

[4] Visual perception of liquids: Insights from deep neural networks
JJR van Assen, S Nishida, RW Fleming
PLoS computational biology 16 (8), e1008018, 2019

[5] Contrast sensitivity functions in autoencoders
Q Li, A Gomez-Villa, M Bertalmío, J Malo
Journal of Vision 22 (6), 8-8, 2022

[6] Convolutional neural networks are not invariant to translation, but they can learn to be
V Biscione, JS Bowers, JMLR 2021

**Summary Of The Paper:**

The authors present an study in order to check whether an standard CNN (Resnet-50) or an standard CNN in which both a foveated rendering from the retina and a log-polar representation have benn added behaves as humans in the image inversion effects. In order to do that they perform a variaty of experiments on different datasets, from faces, to dogs, cars, and a general one.

**Summary Of The Review:**

The paper is of interest and is nicely written. However, I believe it is still not at the level for a conference like ICLR.

---

### Official Review · Reviewer_8Y2A · 2022-10-24

**Confidence:** 5
**Clarity, Quality, Novelty And Reproducibility:** See Strength And Weaknesses
**Correctness:** 3
**Technical Novelty And Significance:** 3
**Empirical Novelty And Significance:** 3
**Recommendation:** 5

**Strength And Weaknesses:**

Pros:
1. The motivation is clear.
2. The paper is well-written and organized.
Cons:
1. The main contributions are not clear.
2. Some related works are missing, e.g., 3D Face Reconstruction from A Single Image Assisted by 2D Face Images in the Wild.

**Summary Of The Paper:**

The authors explored image inversion effects and the impact of visual expertise on performance. By using images of faces, objects, cars, and dogs, the authors were able to show that LPNet, a convolutional neural network that includes a foveated retina and the log polar mapping from the visual field to V1, can reproduce experimental results of image inversion despite being nominally rotation invariant.

**Summary Of The Review:**

See Strength And Weaknesses

---

### Official Review · Reviewer_yrmA · 2022-10-24

**Confidence:** 4
**Correctness:** 3
**Technical Novelty And Significance:** 2
**Empirical Novelty And Significance:** 2
**Recommendation:** 5

**Clarity, Quality, Novelty And Reproducibility:**

The paper is generally clearly written and describes the experimental design well.

One key argument in the paper was unclear to me: that a log-polar representation, when fed into a CNN provides rotation and scale invariance. I cannot understand why this is the case from the explanations given: A CNN provides translation invariance, and rotation and scale robustness is only achieved through training on augmented data. In a V1-like log-polar representation, the scale and orientation of a pattern's encoding is dependent on its excentricity, therefore translation in the image leads to a change in scale and orientation. I am not clear how this equates to invariance though (beside the fact that cropping-based data augmentation will implicitly train for rotation+scale robustness): a rotated picture will have different encoding than the original. This should be better explained as this is central to the explanation of the experimental results.

Another aspect that would warrant more explanation is the rather large variance seen in some of the graphs

The LPNet model could have been described in more specifics: I have a fairly clear idea on what it does, but would probably not be able to reimplement it exactly.

The paper uses a custom dataset for faces, that was not provided at reviewing time. For the other visual classes the dataset is a subset of ImageNet, but the indices of those images are unknown.

Providing the code and all stimuli for the experiments would be essential to ensure reproducibility.

One minor issue: one reference is missing in page 9 (ie, [REF])

**Strength And Weaknesses:**

### Strengths
- The paper is generally clear and well written, with a clear description of the  experimental design.
- The hypotheses are well framed with respect to the literature on the topic
- The experimental setup is generally clear and well chosen.
- Effects such as the inversion effect are interesting to study as they could link the high performance of the human visual system to specialisation in processing.

### Weaknesses
- Relevance to ICLR: I am not fully convinced that ICLR is the right venue for this work. Although the methodology involves neural networks, the hypothesis and conclusion are relevant to the human visual system and psychology in particular. Due to the experimental choices (mainly, artificially limited data augmentation), the results have limited bearing to the study of artificial neural networks or computer vision in general.
- Some of the experimental results seem fairly obvious: A CNN trained with limited rotation fails at recognising examples under large rotation/inversion. A network that had some additional rotation training due to design (log-polar mapping) performs better.
- The central argument that the inversion effect is due to the fact that recognition is mediated by configurations that are disrupted by inversion is not really supported by the experimental setup or results: the configurations learnt by a CNN are not robust to rotation if only limited rotations were included in the training data. This is a consequence of known limitations of CNN and of experimental design, but what does it say about humans? We see faces in much a wider variety of poses than suggested by the dataset?
- Is the conclusion any deeper than: learning systems can recognise patters in similar conditions than the ones they were exposed to during training?
- In Experiment III, it is not clear how large each "set of images" is. It would be good also to have a measure of variance on those numbers, as I expect it may be quite large depending on the face configuration.
- The argument about visual expertise could be refined in my view. In Figure 4, the loss in accuracy due to inversion climbs up to 20 classes and to a lesser extent up to 60, but seems fairly stable after that. It is unclear why visual expertise would plateau in this manner (error bars are also extremely large on the LPNet graph).

**Summary Of The Paper:**

This article is interested in the so-called "Inversion effect" in human vision: the fact that although humans are as a rule excellent at recognising and discriminating faces, performance falls to very low levels when faces are seen upside down. This is an interesting effect as we are generally apt at recognising objects in various poses.

The authors' two hypotheses are that i) the log-polar mapping of the early visual system (retina+V1) plays a role in this effect (this is implicit in the paper's methodology); and 2) that this effect is explained by visual expertise: ie, that face recognition is a highly specialised competency that depends on complex configurations that are disrupted when a face is seen upside-down.

The paper's hypotheses are examined by recognition experiments using standard CNNs and a proposed log-polar neural network (so-called LPNet), on recognition tasks using faces and other objects as a baseline. Visual expertise is modeled as the number of classes the networks have to discriminate.

Results show that:
- validation performance drops with the number of classes in all cases.
- validation performance drops faster for inverted stimuli
- performance on inverted stimuli tends to be worse for the CNN compared to LPNet
- CNNs appear to be more reliant on global configurations than LPNet

**Summary Of The Review:**

In summary, I find the research idea interesting, but I have a number of concerns about the paper as it stands:
- Although the research question and hypotheses are interesting, it is not fully clear to me that the experiments really validate the hypotheses, and that the contrived experimental setup allows for conclusions that are not already well known.
- To justify presentation at ICLR, it would be important that the authors clarify what are the contributions and implications of their findings for computer vision and neural networks research. If the main contribution is psychological in nature maybe this is not the best venue for this paper.
- Code and data would need to be made available to allow reproduction of the results.

---

> ### Author Response · Authors · 2022-11-05
> **Just a quick response to your confusion about invariances**
>
> "One key argument in the paper was unclear to me: that a log-polar representation, when fed into a CNN provides rotation and scale invariance. I cannot understand why this is the case from the explanations given: A CNN provides translation invariance,"
>
> In the log-polar plane, scale is just a shift left or right, while rotation in the image plane is a shift up and down. A relatively accessible article on this is https://vislab.isr.tecnico.ulisboa.pt/wp-content/uploads/2012/12/10-ras-traver.pdf See Figure 3 of that paper. In Hinton's terminology, this is $equivariance$, which he uses to describe what happens in convolutional networks feature maps when something is shifted in the image plane - the feature map shifts in the same way. In convnets, you get translation $invariance$ by the combination of equivariance plus pooling, which gives the same output when the shift is small enough that the max feature value is the same. Now, since a convnet is (relatively) translation invariant, when the input is scale and rotation equivariant, that plus translation invariance gives you scale and rotation $invariance$. However, because a shift in the fixation point on the image gives a completely different log-polar representation, translation invariance (with respect to translations of the original image) is lost. Hence you need to sample the image at many fixation points, and learn that all of those represent the same object through the magic of gradient descent.
>
> Does that make sense?

---

> > ### Comment · Reviewer_yrmA · 2022-11-25
> > **About invariance**
> >
> > Thank you for responding to this point and for the reference. I understand your argument, but I still think that calling this property invariance is somewhat misleading: if you process an image containing a noncentral object, then a second picture where the object in question has been scaled or rotated, the encoding will be completely different, not shifted. It is only for a limited class of rotation/scaling operations that the invariance property is observed (eg, centered rotations). In the paper you cite I note that invariance is in inverted commas, with a note of the limitations.
> >
> > A clarification in the text of which transformations the model is invariant to (centered rotations & scaling vs general rotation and scaling) and which are (hopefully) learnt during training would be helpful to the discussion in my view.

---

### Official Review · Reviewer_rdQQ · 2022-10-25

**Confidence:** 5
**Correctness:** 2
**Technical Novelty And Significance:** 2
**Empirical Novelty And Significance:** 3
**Recommendation:** 5

**Clarity, Quality, Novelty And Reproducibility:**

The paper starts of quite clear in the first 2-3 pages, but later has a sense of being rushed for the last 2 pages (there are even some typos of "[REF]") in the manuscript.

The reproducibility aspect however seems very reasonable, and it should be able to do so with the details provided in the paper. Kudos as well to the authors for using error bars in these experiments!

**Strength And Weaknesses:**

See below for Summary of Review. TLDR: I think the main strength of this paper is the scientific question and trying to understand what are the computational mechanisms of human perception that when simulated can disrupt face perception. The main weakness of this paper is that I do not understand the premise of using a foveated field of view to account for holistic processing of faces when faces can be accurately identified when they also lie within our fovea (!)

**Summary Of The Paper:**

Through of a set of "machine psychophysics" experiments, the authors probe on the image inversion effects comparing faces and other objects with a family of neural networks that are "classical", and other that incorporate both foveation + log-polar mapping to simulate visual processes up to V1. Authors find that their log-polar + foveation neural network model explains the image inversion effect while modern CNN's do not.

**Summary Of The Review:**

I think the topic of this paper is quite exciting as it can greatly advance out understanding of face perception in humans, and overall let us know what makes faces so special compared to everyday objects (and this could in turn be helpful for machine vision).

However: while the experiments in this paper are done with great precision and detail (error bars, optimizers are well documented to train the networks presumably trained from scratch, multiple classes & datasets), I think the premise of explaining the image inversion effect through early staged of visual processing is quite bold, (but exciting!), however I do not know what foveation has to do with this. If humans can recognize faces without foveation, then how does that results fit into this methodology? I could argue the same for log-polar mapping.

Perhaps a missing experiment that remains to be done here is to re-do the experiments with the faces shrunken to the foveal area of the log-polar mapping and later evaluate the outcomes? Quite frankly, I'm also a bit confused, as I'm not sure how to make sense of these results even by having the log-polar network as a control. Maybe there is a confounding variable in this analysis, but I can't wrap my head around what the confounding variable is or could be.

It occurs to me that maybe some additional experiments could be done with scenes, but I'm not sure what is to be proven: perhaps that there is no image inversion effects in scenes? (I'm arguing for scenes here because that is a believable scenario where I can see having a model that incorporates a wide field of view through a log-polar transform be plausible -- similar to Deza & Konkle, ArXiv 2020.)

Missing references:
* Brain-like functional specialization emerges spontaneously in deep neural networks, by Dobbs et al. Science Advances 2022.
(In general several works from Dobbs that involve training neural networks on faces and objects and comparing these resuilts)

---

### Decision · Program_Chairs · 2023-01-20

**Decision:**

Reject

**Justification For Why Not Higher Score:**

General consensus from the reviewers that the paper should be rejected. No rebuttal provided.

**Justification For Why Not Lower Score:**

NA

**Metareview: Summary, Strengths And Weaknesses:**

This paper tested whether modern CNNs with and without cortical magnification/foveation exhibit an "inversion effect" as humans do. The results presented find that cortical magnification/foveation is needed to account for the phenomenon. Criticisms were brought up by the reviewers (includingthe need to clarify some of the experiments and providing a rational for a plausible connection between the inversion effect and the need for cortical magnification/foveation) which the authors did not attempt to address.